# The Role of Coronary CT Angiography in the Evaluation of Dual Left Anterior Descending Artery Prevalence and Subtypes: A Retrospective Multicenter Study

**DOI:** 10.3390/jpm13071127

**Published:** 2023-07-12

**Authors:** Nicola Maggialetti, Sara Greco, Giovanni Lorusso, Cristiana Mileti, Gabriella Sfregola, Maria Chiara Brunese, Marcello Zappia, Maria Paola Belfiore, Pasquale Sullo, Alfonso Reginelli, Nicola Maria Lucarelli, Arnaldo Scardapane

**Affiliations:** 1Interdisciplinary Department of Medicine, Section of Radiology and Radiation Oncology, University of Bari “Aldo Moro”, 70124 Bari, Italy; n.maggialetti@gmail.com (N.M.); saragrecosa@gmail.com (S.G.); cristianamileti@gmail.com (C.M.); gabriella.sfregola@gmail.com (G.S.); lucarelli.nico@gmail.com (N.M.L.); arnaldo.scardapane@uniba.it (A.S.); 2Diagnostic Imaging Section, Department of Medical and Surgical Sciences & Neurosciences, University of Molise, 86100 Campobasso, Italy; mariachiarabrunese@gmail.com; 3Department of Medicine and Health Science, University of Molise, 86100 Campobasso, Italy; marcello.zappia@unimol.it; 4Department of Precision Medicine, University of Campania “Luigi Vanvitelli”, 80123 Naples, Italy; mariapaola.belfiore@unicampania.it (M.P.B.); alfonso.reginelli@unina2.it (A.R.); 5Sant’Anna e San Sebastiano Hospital of Caserta, Radiology Division, 81100 Caserta, Italy; pasqualesullo@gmail.com

**Keywords:** coronary computed tomography angiography (CCTA), coronary artery, dual left anterior descending artery (LAD), myocardial bridge, angina

## Abstract

Background: The aim of this multicenter study was to evaluate the prevalence and features of dual left anterior descending artery (LAD) subtypes using coronary CT angiography (CCTA). Methods: A retrospective multicenter analysis of 2083 CCTA from December 2020 to November 2022 was conducted to search for the presence and morphological features of dual LAD. The two classifications used were the updated classification of Spindola-Franco and the Jariwala classification. Statistical tests were conducted to evaluate the prevalence of dual LADs among sexes and its association with angina in patients without significant coronary stenoses and/or associated cardiac anomalies. Results: Dual LAD was observed in 124 (5.96%) patients analyzed. According to the Spindola-Franco revisited classification, type I dual LAD was the most common (71/124, 57.26%). According to the Jariwala classification, all cases were group I. In the general population, there was a higher prevalence of dual LAD among females (7.3% females vs. 5.1% males; *p* value: 0.04). No statistically significant difference was found in the prevalence of angina in the dual LAD population compared to the no dual LAD population (2.1% vs. 1.5%; *p* value: 0.10). Conclusions: The acknowledgment and reporting of LAD duplication is helpful for an optimal management of coronary patients with this condition. Dual LAD was more frequent in the female population, mainly not related with angina. Myocardial bridge was more frequent in the dual LAD population than in the no dual LAD population.

## 1. Introduction

Coronary artery anomalies (CAAs) are a group of congenital conditions characterized by abnormal origin, course or termination of any of the three main epicardial coronary arteries. Most of these anomalies are asymptomatic and discovered as incidental findings during cardiac imaging [1] but sometimes they might present with chest pain, ECG anomalies or with life-threatening complications like acute myocardial infarction or sudden cardiac death (SCD) [2].

Coronary computed tomography angiography (CCTA) moved into the diagnostic realm with the introduction of multi-detector row CT and the development of electrocardiographically (ECG)-synchronized scanning and reconstruction techniques.

Thanks to the widespread use of invasive and noninvasive coronary imaging, the recognition of CAAs has increased.

In the general population, CAAs are encountered in 0.7% to 18.4% on CCTA studies [3].

For several decades, diagnosis of CAAs has been performed with coronary catheter angiography (CCA), but nowadays ECG-gated multi-detector row CCTA is considered superior for identification and characterization of coronary anomalies, thanks to recent advancements in software and hardware, and excellent temporal and spatial resolution [4,5]. It has been also reported that CCTA may be superior to CCA in defining the ostial origin and the proximal path of anomalous coronary branches; also, it should not be forgotten that CCTA has the advantages of being noninvasive, safer and faster than CCA [6,7].

Among the coronary artery anomalies, we focused on course anomalies and particularly on the duplication of the LAD artery.

The LAD artery arises from the left main coronary artery (LMCA), passes to the left of the pulmonary trunk and turns anteriorly to course in the anterior interventricular groove (AIVG) toward the apex. It provides the diagonal branches to the anterior free wall of the left ventricle and the septal branches to the anterior interventricular septum [8]. Duplication of the LAD is a rare anatomical variant characterized by two coronary branches known as the “short LAD” and “long LAD”, that run in or approximate to the AIVG [9]. Duplication of the LAD artery on CCTA has been reported to occur in 0.68 to 6% of the general population in different case series [10]. The prevalence of dual LAD described in the CCA literature was 1%, which is lower than the prevalence described for CCTA [11,12,13]. CCA may suffice to establish a diagnosis when both short and long LAD arises from common LAD (LAD proper), because when the long LAD originates from the opposite coronary sinus or when the vessel is totally occluded there is a risk of misrecognition. Furthermore, although mortality associated with LAD occlusion should be minimized if there are two LADs supplying the same territory of a single LAD, a total occlusion of a long LAD could be not highlighted during CAA, producing a false negative.

Duplication of the LAD artery running on the left side of the ventricle should not be confused with a LAD artery and a diagonal branch running parallel to each other. Such a parallel diagonal branch does not reenter the AIVG [14].

The first mention of a dual LAD in the literature was described by Waterson et al. in 1939 [15]. In 1983, Spindola-Franco et al. published the first review based on CCA of four types of dual LAD anomalies, which provided an angiographic description and a primarily numerical method of classification [11].

To date, since the initial classification by Spindola-Franco, the dual LAD variants number has increased up to 13 types. The latest subtype was identified in April 2021 by Pellegrini et al. [16].

In 2020, Jariwala P. et al. reviewed the current state of the art and proposed a novel classification of dual LAD in order to create a uniformity in the identification and diagnosis of the dual LAD system [14].

The aim of our study was to identify the morphological features and the prevalence of dual LAD using CCTA, in order to raise awareness of this underestimated variant among radiologists. Other aims were to evaluate the dual LADs’ prevalence across gender and to examine the association of dual LAD with angina in patients without significant coronary stenoses and/or associated cardiac anomalies. Finally, we evaluated the prevalence of myocardial bridge in the dual LAD population compared with the no dual LAD population.

## 2. Materials and Methods

### 2.1. Study Population

The study draft was approved by our institutional review board. A retrospective multicenter analysis of 2083 consecutive CT scans of all patients (male 1319, female 764; mean age 65.90; DS 12.21) undergoing CCTAs from December 2020 to November 2022, was conducted to identify the morphological features and prevalence of dual LAD. The patients were recruited from the “Azienda Ospedaliera-Universitaria Consorziale Policlinico di Bari” (Italy); “Università degli studi del Molise- Fondazione Potito Istituto di Ricerca Diagnostica per Immagini” (Campobasso, Italy) and the “Università degli studi della Campania “Luigi Vanvitelli”.

These examinations were re-evaluated by 4 radiologists blinded about clinical information and radiologic reports; disagreements were resolved by open discussion. In our study, the patient inclusion criteria for study were high quality images with an adequate signal-to-noise ratio, a correct vascular enhancement and a motion-free; 3 exams with inadequate images have been excluded.

Various indications for CCTA included suspected coronaropathy diseases, preoperative work up of patients with valvular heart disease, coronary evaluation in patients with left ventricular dysfunction or vasculitis, follow-up after CABG (coronary artery bypass grafting) or PCI (percutaneous coronary intervention) and screening tests.

For all patients, we collected the following data: clinical indication for CCTA, gender, age, dual LAD presence/absence, significant coronary stenosis, associated anatomic variants and patient medical history.

### 2.2. CT Scans

Out of 2083 ECG-gated CCTA, 530 were obtained with Aquilon Premium (CANON Medical System, Tustin, CA, USA) from “U.O. Radiodiagnostica Universitaria di Bari”; 583 CCTA examinations were performed using Revolution CT © 2023 General Electric Company from “Università degli studi della Campania “Luigi Vanvitelli”; and the last 970 CT scans were acquired with Aquilion ONE GENESIS Edition, Canon Medical Systems, Tokyo, Japan from “Università degli studi del Molise- Fondazione Potito Istituto di Ricerca Diagnostica per Immagini” (Campobasso, Italy).

The acquired CT data were collected through our institutional PACS (Carestream Health, Rochester, NY, USA; Enterprise Imaging, Biesse Medica, Roma, Italy; Suitestensa EBit, Genova, Italy). All CCTA exams were performed with ECG-gating prospectively or retrospectively, according to the patient’s heart rate.

Images were analyzed using tridimensional algorithms, such as maximum intensity projection (MIP) and volume rendering (VR), that allow angiographic-like visualization of the coronary arteries and multiplanar reformats (MPR).

### 2.3. Image Evaluation

Dual LAD subtypes were classified according to both Spindola-Franco extended classification (Table 1) and Jariwala P. novel classification (Figure 1).

In Jariwala’s classification, dual LAD is categorized into one of the three groups based on the origins of the two LADs; those that originate entirely from the left coronary sinus (Group I), those that originate partially from both the left and right coronary sinuses (Group II), and those that originate entirely from the right coronary sinus (Group III). It is further subgrouped based on their four courses: epicardial or prepulmonic or anterior (A); interarterial or between (B); retroaortic or posterior (P); and intramyocardial or septal (S) [17]. In Spindola-Franco’s classification, there is no subdivision into groups and subgroups as in the Jariwala’s classification but each duplication is classified into types based on the basis of course and origin.

The other anatomical variants have been referenced according to the “Coronary Artery Anomalies: Classification and ECG-gated Multi–Detector Row CT Findings with Angiographic Correlation” of Radiographics [4].

### 2.4. Statistical Analysis

Statistical analysis was performed with SPSS software (version 26.0 SPSS Inc., Chicago, IL, USA).

Descriptive statistics, including means, standard deviations and percentages, were used to summarize the data.

Chi-square test was used to compare the prevalence of dual LAD between male and female populations. Furthermore, chi-square test was performed to compare the prevalence of angina and myocardial bridge between the dual and no dual populations. *p* values less than 0.05 were considered statistically significant.

## 3. Results

A total of 2083 CCTAs were enrolled in this study; 3 patients were excluded due to poor exam quality. Clinical characteristics of all patients are presented in Table 2.

Dual LAD was detected in 124/2080 patients (5.96%) (Figure 2); 56/124 (45.16%) patients were female, 68/124 (54.84%) were male, and mean age was 65 (DS 11). Stable ischemic heart disease was the most common indication for CCTA and hypertension was the most common risk factor in patients with dual LAD.

According to the Spindola-Franco updated classification, type I dual LAD (Figure 3) was the most common, being observed in 71/124 cases (57.26%). Type II dual LAD (Figure 4) was observed in 11/124 cases (8.87%), type III (Figure 5) in 9/124 (7.26%), and type XIII in 33/124 (26.61%) (Figure 6).

Considering the Jariwala classification, all cases are group I; 115/124 (92.74%) subgroup A (Figure 3, Figure 4 and Figure 5) and 9/124 (7.26%) subgroup S (Figure 5).

Some extra-cardiac and vascular abnormalities and variations were associated with dual LAD variants.

Right dominant circulation was the most common pattern of arterial dominance, seen in 110/124 cases (88.71%). Co-dominant circulation was seen in 2/124 cases (1.61%), whereas left dominance was seen in 12/124 cases (9.68%).

Myocardial bridges were observed in 15/124 cases (12.10%).

The other vascular variants associated with dual LAD were 1/124 anomalous circumflex artery (CX) with retroaortic course arising from RCS, 1/124 bicuspid aortic valve, 1/124 right coronary artery (RCA) high takeoff, 1/124 anomalous origin of RCA from the LCS, 1/124 CX origin from RCA, 1/124 trifurcation of the left main coronary artery (LMCA) and 1/124 CX aplasia.

Significant atherosclerotic stenosis (>50% diameter stenosis) was seen in 22/124 (17.74%) patients with dual LAD: single vessel disease was the most common pattern and was seen in 13/22 patients (59.9%), double vessel disease was seen in 2/22 patients (9.09%), while triple vessel disease was seen in 7/22 patients (31.82%) (Table 3).

Prevalence of dual LAD was evaluated between male and female population. In the general population, there was a statistically significant difference in the prevalence of dual LAD between sexes (56/764 female 7.3% vs. 68/1316 5.1% male; p:0.04).

Prevalence of angina events was evaluated among the population without significant stenosis or associated cardiac anomalies. In the dual LAD population, there was an incidence of angina of 2.1% (20/92) while in the general population there was an incidence of 1.5% (220/1434). Chi-square test showed no significant difference between the two groups (*p* value: 0.10).

## 4. Discussion

The prevalence of dual LAD was 5.96% (124 dual LADs among 2080 cases), which was the largest series in the existing literature using CCTA. Other studies with smaller series presented a prevalence of 1.3% (25 dual LADs among 1912 cases) [17] and 4% (56 dual LADs among 1337 cases), respectively [10].

The prevalence of dual LAD reported in this study based on CCTA is about five times higher than the prevalence described in the CCA literature (5.96% and 1%, respectively) [10,11].

CCTA has become the method of choice for the detection and classification of congenital coronary artery anomalies. As compared to CCA, cardiac CT allows to define the origin, course and termination of the coronary artery with detailed high-quality and three-dimensional images [3,10,11,18,19]. CCTA is also noninvasive, faster, and safer that CCA and also is dose-reduction technique [10].

In this study, dual LADs were classified according to both the Spindola-Franco extended classification and the Jariwala novel classification for the first time. The first one is mostly descriptive and there is no subdivision into groups and subgroups but each type has its own origin and course, while the second one is divided into groups based on the origin and subgroups based on the course; in this way, Jariwala’s classification guarantees a uniformity in the identification and diagnosis of dual LADs.

According to the existing literature, among the various subtypes of dual LADs in the Spindola-Franco revisited classification, type I is the most common subtype [4]. In line with the literature, in this study, 71/124 cases (57.26%) of type I were found. In order of frequency, type XIII (new subtype) was found in 33/124 (26.61%), which is more than the other subtypes. Type XIII was described as two long LADs which both leave the AIS and course out to the apex, so despite the dual LAD’s definition, it is no longer possible to recognize a short and a long LAD. One of the vessels courses laterally and the other courses medially of the AIS. Based on Jariwala’s classification, all cases are group I; 115/124 (92.74%) subgroup A and 9/124 (7.26%) subgroup S.

Dual LAD could be associated with other coronary artery features, particularly with left dominance (12/124 cases, 9.68%) and myocardial bridge (15/124 cases, 12.10%). The prevalence of myocardial bridge in the dual LAD population (15/124, 12.10%) was in line with the prevalence described in the literature in the general population (2–15%) [20]. In our study, no dual LAD population presented myocardial bridge in 113/1956 cases (5.7%); a higher prevalence of myocardial bridge in dual LAD (15/124) population than in no dual LAD population (113/1956) was observed (*p*-value: 0.01).

Also, a statistically significant difference based on gender was found in the dual LAD population, with a higher prevalence in females (56/124, 45.16%) than males (68/124, 54.84%).

Approximately 80% of the dual LAD anomalies are asymptomatic and represent incidental findings in exams performed for the most common indication for CCTA, such as stable ischemic heart disease and hypertension; just 20% of dual LAD were associated with symptoms [21,22]. The study showed 36/124 (29%) symptomatic dual LAD, 35/124 (28.22%) with stable angina and only 1/124 (0.8%) with unstable angina. Among dual LAD with angina, 19/124 (15.32%) patients had no other cardiac anomalies that could explain the symptoms. The prevalence of angina in dual LAD population without significant stenosis or cardiac anomalies was compared with the prevalence of no dual LAD population with the same characteristics and no significant statistical difference was found. These results, in line with the previous literature, confirmed the mainly asymptomatic nature of the dual LAD types found in this study, unless it was associated with significant anatomical cardiac variants or significant coronary stenosis [1,21,22]. The aberrant artery origin from the opposite coronary sinus or the RCA and subsequent course may cause symptoms [9,11,19,22,23,24]. Although none of these cases have been found.

A coronary artery arising from the opposite or noncoronary sinus could take different courses depending on the anatomic relationship between the anomalous vessel, aorta and pulmonary trunk. The four common courses are interarterial (between the aorta and the pulmonary artery), retroaortic, prepulmonic or septal (subpulmonic) [25]. It is important to know which course is taken by the two LAD; while retroaortic, prepulmonic and septal courses seem to be benign, an interatrial course is associated with a SCD, hence, there is an indication for surgical correction in case of evidence of myocardial ischemia or previous syncope [26,27,28,29]. During a CCA, there is a risk of mistaking a coronary anomaly, especially when the long LAD originates from the opposite coronary sinus, and this affects the result of the treatment. To date, it is not clear if there is a relationship between coronary atherosclerosis and dual LAD; in fact, due to the high prevalence of coronary atherosclerosis, these conditions may coexist [9,10,11,19,30].

In this study, a significant atherosclerosis (>50% diameter stenosis) was found in 19/124 (15.32%) cases. The limits of the study were related to its multicentric nature: each center used a different CT scanner and acquisition protocols. Despite the size of the study, not all dual LAD types were observed due to the rarity of these variants [10,11,19]. However, the possibility of finding these anomalies cannot be ruled out if the sample size would have been larger.

In conclusion, although dual LAD anomalies are described as a rare congenital coronary, anomaly is not so rare as confirmed by this study, in which a prevalence of 6% was found. According to our study, the prevalence of dual LAD in the general population was statistically higher in the female population than males.

## 5. Conclusions

CCTA represents an essential imaging tool for the assessment of coronary anomalies. Although dual LAD is mainly asymptomatic and not related with angina, it is important for radiologists to have knowledge of these anomalies and report them if they have an aberrant origin, a total occluded vessel or a severe stenosis, in order to prevent misunderstanding on coronary angiography and inadequate therapeutic approach.

## Figures and Tables

**Figure 1 jpm-13-01127-f001:**
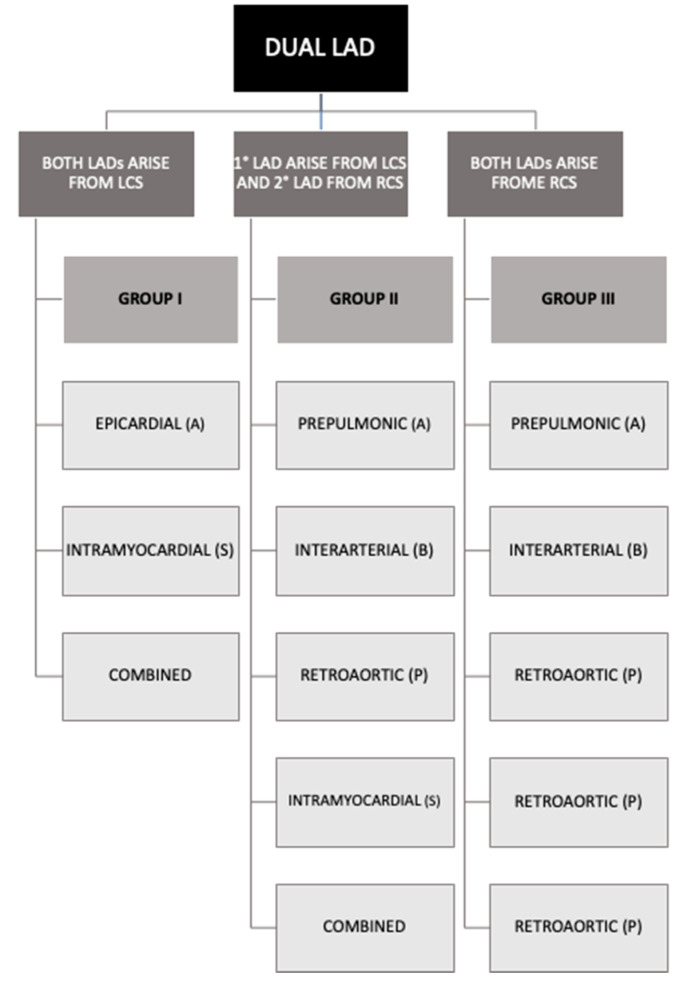
Scheme of Jariwala’s novel classification of dual LAD [16]. LAD: left anterior descending artery; LCS: left coronary sinus; RCS: right coronary sinus.

**Figure 2 jpm-13-01127-f002:**
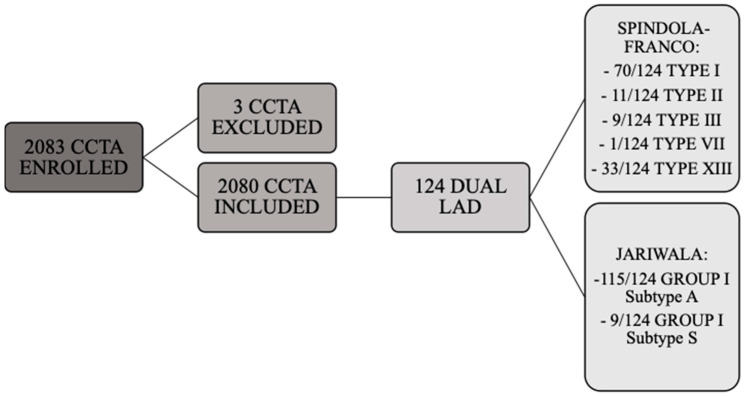
Flowchart of study enrollment and dual LAD subtypes.

**Figure 3 jpm-13-01127-f003:**
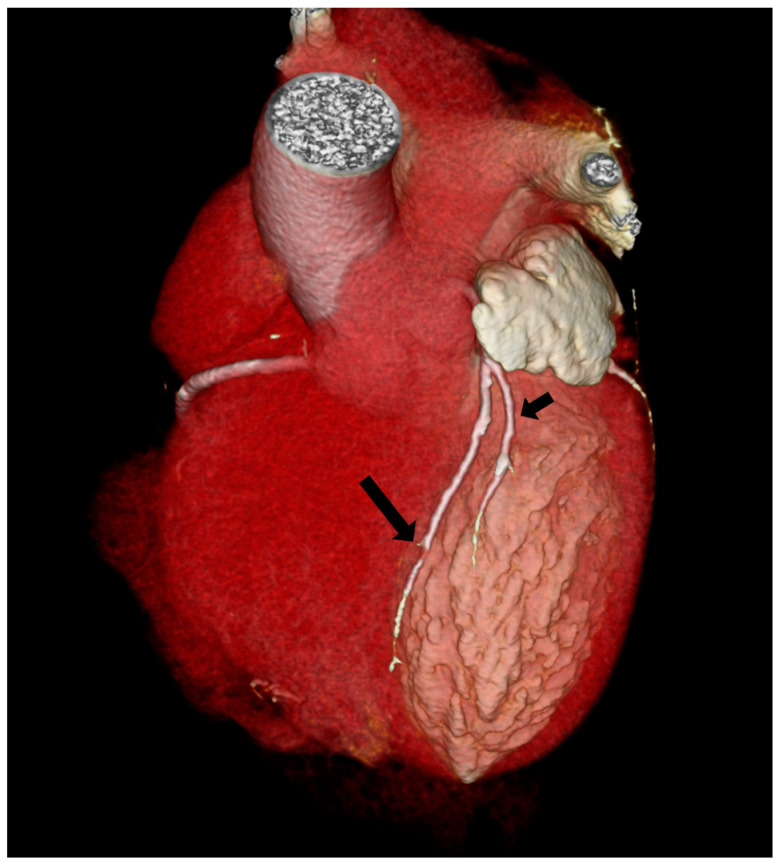
Type I Spindola-Franco and group I subgroup A Jariwala dual left anterior descending artery (LAD) in a 74-year-old male patient. The 3D volume rendered reconstruction shows a short arrow, that identify short LAD, and a long arrow, that identify long LAD; both LAD originating from the LAD proper. Long LAD descends on the LV side of the proximal AIS and enters the distal anterior in-terventricular sulcus.

**Figure 4 jpm-13-01127-f004:**
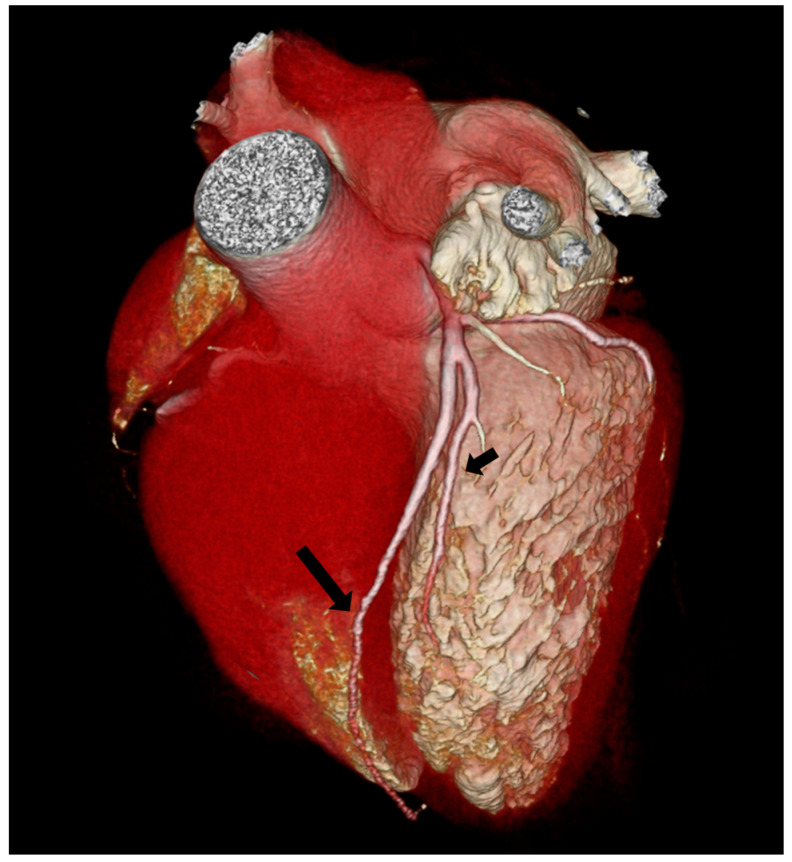
Type II Spindola-Franco and group I subgroup A Jariwala dual left anterior descending artery (LAD) in a 48-year-old male patient. The 3D volume rendered reconstruction shows a short arrow, that identify short LAD, and a long arrow, that identify long LAD; both LAD originating from the LAD proper. Long LAD descends on the RV side of the proximal AIS and enters the distal anterior interventricular sulcus.

**Figure 5 jpm-13-01127-f005:**
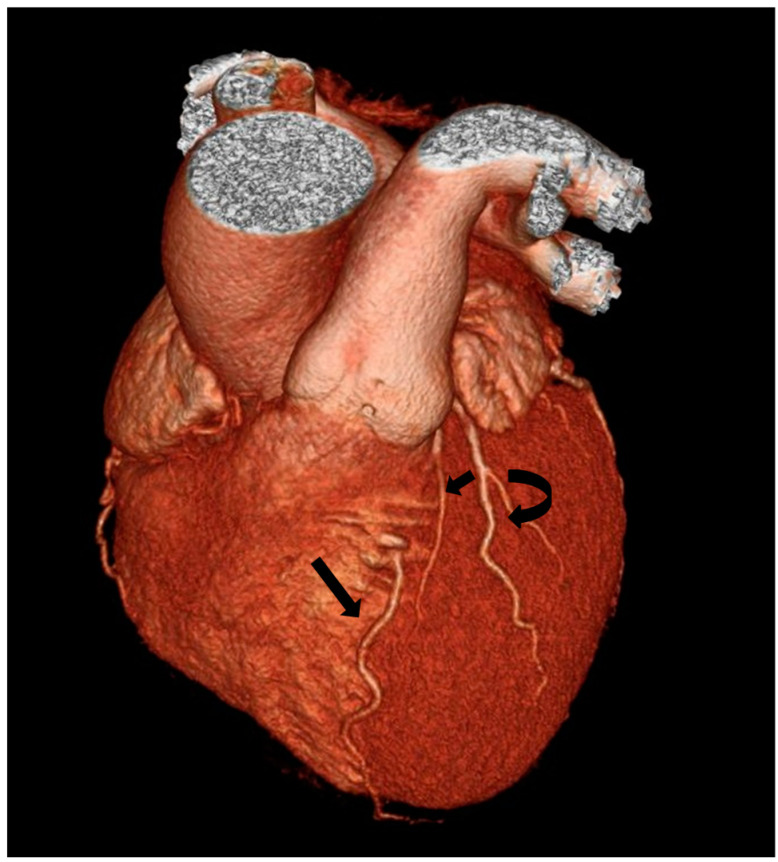
Type III Spindola-Franco and group I subgroup S Jariwala dual left anterior descending artery (LAD) in a 60-year-old male patient. The 3D volume rendered reconstruction shows a short arrow, that identify short LAD, and a long arrow, that identify long LAD; both LAD are coursing in the AIVG. The long LAD is on the LV side of the AIVG and has an intramyocardial course. The proper LAD gives rise to a left ventricular diagonal branch, that is identified with the curved arrow.

**Figure 6 jpm-13-01127-f006:**
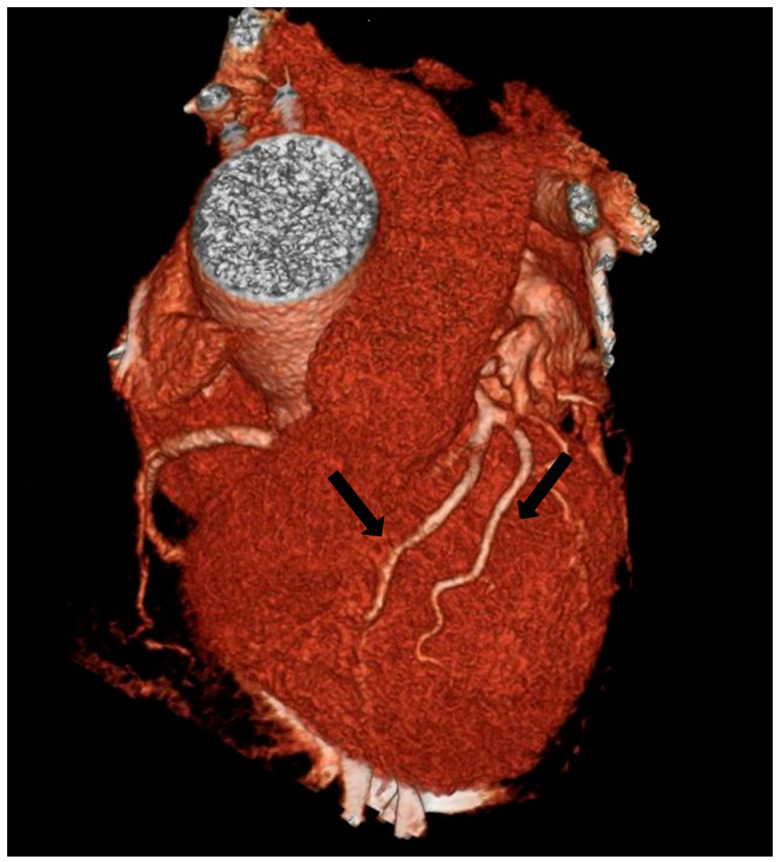
Type XIII Spindola-Franco and group I subgroup A Jariwala dual left anterior descending artery (LAD) in a 63-year-old female patient. The 3D volume rendered reconstruction shows two long LADs which course out to the apex, each one identified with a long arrow.

**Table 1 jpm-13-01127-t001:** Spindola-Franco extended classification of dual LAD variants [15]. AIS: anterior interventricular sulcus. LAD: left anterior descending artery. LMCA: left main coronary artery. LCS: left coronary sinus. LV: left ventricle. RCA: right coronary artery. RV: right ventricle. RCS: right coronary sinus. RVOT: right ventricle outflow tract. PA: pulmonary artery.

	Short LAD	Long LAD
I	Originates from the LAD proper and terminates in the proximal AIS	Originates from the LAD proper, descends on LV side of the proximal AIS and terminates in the distal AIS
II	Originates from the LAD proper and terminates in the proximal AIS	Originates from the LAD proper, descends on RV side of the proximal AIS and terminates in the distal AIS
III	Originates from the LAD proper and terminates in the proximal AIS	Originates from the LAD proper and terminates in the proximal AIS or in the apical septum
IV	Originates from the LMCA and terminates in the proximal AIS	Originates from RCA. courses along an anomalous prepulmonic course anterior to RVOT and reenters the distal AIS
V	Originates from the LCS and terminates in the proximal AIS	Originates from RCS, courses along an intramyocardial course and emerges epicadilly to enter the distal AIS
VI	Originates from the LMCA and terminates in the proximal AIS	Originates from the RCA, courses between the RVOT and the aortic root and emerges in distal AIS
VII	Originates from the LAD proper and terminates in the proximal AIS	Originates from the LAD proper, courses along the LV side of proximal AIS and terminates in the distal AIS
VIII	Originates from the LMCA and terminates in the proximal AIS	Originates from the Mid-RCA, courses along the RV and traverses the apex, terminating in the distal AIS
IX	Originates from the LAD proper and terminates in the proximal AIS	Originates from the LAD proper, courses along the LV side of the AIS, enters the distal AIS and terminates before the apex
X	Originates from the LMCA and terminates in the proximal AIS	Originates from RCS, courses along an anomalous prepulmonic course anterior to RVOT and reenters the distal AIS
XI	Originates from RCS, takes an intramyocardial course and terminates in the proximal AIS	Originates from RCS, courses along an anomalous prepulmonic course anterior to RVOT and reenters the distal AIS
XII	Originates from the LMCA (that originates from RCS) and terminates in the proximal AIS	Originates from the RCS, courses anterior to the main PA and terminates in the distal AIS
XIII	/	Two long LADS, both of which leave the AIS and course out to the apex (one courses laterally so the other courses medially of the AIS)

**Table 2 jpm-13-01127-t002:** Clinical characteristics of enrolled patients. CCTA: coronary computed tomography angiography; CAD: coronary artery disease.

Characteristics	General Population (2080)	Dual LAD (124)	No Dual LAD (1956)
Age, mean (standard deviation)	66.09 (DS 11.9)	65.65 (DS 11.22)	66.12 (DS 11.97)
Female	764 (36.70%)	56 (45.16%)	708 (36.20%)
Male	1316 (63.30%)	68 (54.84%)	1248 (63.80%)
Indication for CCTA:			
Unstable angina	20 (0.96%)	1 (0.81%)	19 (0.97%)
Stable angina	413 (19.86%)	35 (28.23%)	378 (19.33%)
Preoperative work up	47 (2.26%)	4 (3.20%)	43 (2.20%)
Follow-up	283 (13.61%)	9 (7.26%)	274 (14.01%)
Screening test	1299 (62.5%)	75 (60.48%)	1224 (62.58%)
Other causes	18 (0.86%)	0 (0%)	18 (0.86%)
Cardiovascular risk factors:			
Hypertension	1053 (50.63%)	69 (55.65%)	984 (50.31%)
Smoking	112(5.34%)	10 (8.06%)	102 (5.21%)
CAD familiarity	99(4.76%)	12 (9.68%)	87 (4.45%)
Dyslipidemia	29(1.39%)	2 (1.61%)	27 (1.38%)
Significant stenosis (>50% diameter stenosis):			
Present	426 (20.48%)	19 (15.32%)	407 (20.81%)

**Table 3 jpm-13-01127-t003:** Significant CAD atherosclerotic patterns in patients with dual LAD. LAD: L eft ft anterior descending artery; CX: circumflex coronary artery; LMCA: left main coronary artery; RCA: right coronary artery.

Characteristic	Prevalence (%)
Patients with significant stenosis (>50%)	22 (15.32)
Single vessel disease	13 (59.9)
Double vessel disease	2 (9.09)
Triple vessel disease	7 (31.82)
LMCA involvement	2 (10.53)
CX involvement	6 (31.58)
RCA involvement	12 (63.16)
LAD involvement	18 (94.73)

## Data Availability

Interdisciplinary Department of Medicine, Section of Radiology and Radiation Oncology, University of Bari “Aldo Moro”, Bari, Italy.

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
