# Peer review of "The Role of Coronary CT Angiography in the Evaluation of Dual Left Anterior Descending Artery Prevalence and Subtypes: A Retrospective Multicenter Study"

_jpm, 2023, doi:10.3390/jpm13071127_

Round 1
Reviewer 1 Report
- How does Coronary-CT angiography contribute to the evaluation of cardiovascular health?
- The use of different CT scanners and acquisition protocols. This variability in equipment and protocols may introduce potential biases or inconsistencies in the results.
- Some dual left anterior descending artery (LAD) variants. Not all types of dual LAD anomalies were observed in the study due to their infrequency. Therefore, the findings may not fully represent the entire spectrum of dual LAD anomalies.
- The paper focuses on evaluating the prevalence and features of dual left anterior descending artery subtypes using Coronary-CT angiography. It does not discuss any potential areas for future research or improvements in the methodology used in the study.
- How the patients age, amount of contrast used, different protocol parameters affect and associated with the results?
-
- Explain the practical implications of this paper.
Author Response
1. How does Coronary-CT angiography contribute to the evaluation of cardiovascular health?
Response 1: Thanks to recent technological advances, CCTA has improved diagnostic accuracy in detecting coronary artery anomalies and significant coronary stenosis (> 50%) contributing to the assessment of cardiovascular health and of coronary artery course.
Nicol ED, Norgaard BL, Blanke P, Ahmadi A, Weir-McCall J, Horvat PM, Han K, Bax JJ, Leipsic J. The Future of Cardiovascular Computed Tomography: Advanced Analytics and Clinical Insights. JACC Cardiovasc Imaging. 2019 Jun;12(6):1058-1072. doi: 10.1016/j.jcmg.2018.11.037. PMID: 31171259.
2. The use of different CT scanners and acquisition protocols. This variability in equipment and protocols may introduce potential biases or inconsistencies in the results.
Response 2: We are aware of this limit as we wrote in lines 341-342 but the choice of a retrospective multicenter study implicitly involved the use of different scanners and protocols. However, 2080 images included in our study were considered diagnostic to quantify the degree of stenosis and to study the course of the coronary arteries by our four expert radiologists.
3. Some dual left anterior descending artery (LAD) variants. Not all types of dual LAD anomalies were observed in the study due to their infrequency. Therefore, the findings may not fully represent the entire spectrum of dual LAD anomalies.
Response 3: We are aware of this limit as we wrote in lines 342-344, although our sample is quite large we cannot rule out that other variants could be found with an even larger sample.
4. The paper focuses on evaluating the prevalence and features of dual left anterior descending artery subtypes using Coronary-CT angiography. It does not discuss any potential areas for future research or improvements in the methodology used in the study.
Response 4: Our study is mainly descriptive. It would certainly be interesting to investigate the presence of further coronary abnormalities not yet known and probably future technological improvements will contribute to the search for such variants.
5. How the patients age, amount of contrast used, different protocol parameters affect and associated with the results?
Response 5: The age of the patients did not affect the outcome related to the prevalence and subtypes of the dual LAD as it is a congenital condition. The age of patients has an impact on the assessment of CAD, which is prevalent in older people. The amount of contrast and the type of protocol used are instead parameters that could affect the quality of the exam and this as already mentioned above as an intrinsic limit of our multicenter study.
6. Explain the practical implications of this paper.
Response 6: It is useful to know and report the presence and type of duplication of the dual LAD in order to avoid incorrect surgical treatment. During a CCA there is a risk of mistaking a coronary anomaly, especially when the long LAD originates from the opposite coronary sinus or when a dual LAD is totally occluded and not visible at CCA. These situations affect the result of the treatment.
Reviewer 2 Report
The authors of this study were interested in the frequency of a particular coronary anomaly, the dual LAD, on corosCT, in a large retrospective study of more than 2000 patients. They reported a frequency of 6% of this anomaly, anomaly which remains relatively unknown compared to high-risk classic anomalies.
We must congratulate them for this study which required the analysis of a large number of scanners. The impact of the study does not seem to me to be major because, as the authors showed, this anomaly seems benign and was not associated with a greater frequency of anginal pain here (this is the weak point of the paper according to me) but it is the largest study published on this subject, which remains interesting. The comparison of the 2 classifications is also relevant for the paper which is quite pleasant to read. I will make a few more specific remarks:
- Abstract line 38: the conclusion is exaggerated. It cannot be said that the identification of this anomaly is fundamental for optimal management. This is not demonstrated in the paper by the way. To review.
-line 106: how do you define high quality images?. Only 3 scanners excluded out of more than 2000. That seems very few to me. Can you explain these 2 points.
- line 112: cardiovascular risk factors are part of the data collected. To add. The HR plaques are mentioned here and then are no longer found in the results. To be deleted I think. The definition of HRP is given far below (line 147), an error I think.
- line 163: see comment line 106
- Table 2: put a column on the right with the P value for each test
- Table 2, row significant stenosis: remove the row "absent" which is logically deduced
Figures: you use Roman numerals for the classification of Spindola in the text and Arabic numerals for the figures. Homogeize that. Same remark for Jariwala's classification: you speak of "group" in the figure and of "type" in the text.
Figure 5: arrows are missing
Line 236-239: already said in the introduction. I would delete that.
Line 258: 12.10% dual LAD versus 5.7% no LAD: can the authors do a statistical test on that? The difference may be significant ....
Line 288: can you add and compare the calcium scores values between the 2 populations? It is a reflect of the atheromatous burden
Conclusion: for me the important result of the paper is the fairly high frequency of this anomaly. The majority of patients do not go to coronary angiography after a coroscanner which seeks above all to rule out the disease. I'm not sure it can be said that, if you don't mention a double LAD, there will be inadequate theapetutiau. To review.
Author Response
- Abstract line 38: the conclusion is exaggerated. It cannot be said that the identification of this anomaly is fundamental for optimal management. This is not demonstrated in the paper by the way. To review.
It is useful to know and report the presence and the type of duplication of dual LAD in order to avoid incorrect treatment. During a CCA there is a risk of mistaking a coronary anomaly, especially when the long LAD originates from the opposite coronary sinus and in patients with long-standing atherosclerosis, it is possible that both LADs may be occluded or had a significant degree of stenosis; therefore, it is relevant to identify the coronary vessels accurately before CCA.
-line 106: how do you define high quality images?. Only 3 scanners excluded out of more than 2000. That seems very few to me. Can you explain these 2 points.
To define a high quality image we considered an adequate signal-to-noise ratio, a correct vascular enhancement and a motion-free exams. The images were of high quality for the use of advanced technologies and protocols built according to international guidelines.
- line 112: cardiovascular risk factors are part of the data collected. To add. The HR plaques are mentioned here and then are no longer found in the results. To be deleted I think. The definition of HRP is given far below (line 147), an error I think.
Thank you for the suggestion, fixed.
- line 163: see comment line 106
Thank you for the suggestion, fixed.
- Table 2: put a column on the right with the P value for each test
The calculation of p-value is not applicable for the type of data available in this table.
- Table 2, row significant stenosis: remove the row "absent" which is logically deduced
Thank you for the suggestion, fixed.
Figures: you use Roman numerals for the classification of Spindola in the text and Arabic numerals for the figures. Homogeize that. Same remark for Jariwala's classification: you speak of "group" in the figure and of "type" in the text.
Thank you for the suggestion, fixed.
Figure 5: arrows are missing
Thank you for the suggestion, added.
Line 236-239: already said in the introduction. I would delete that.
Thank you for the suggestion, fixed.
Line 258: 12.10% dual LAD versus 5.7% no LAD: can the authors do a statistical test on that? The difference may be significant ....
Thank you for the suggestion, added.
Line 288: can you add and compare the calcium scores values between the 2 populations? It is a reflect of the atheromatous burden
Thanks for the suggestion; it would be an interesting analysis but in our study we didn’t consider the calcium score because in our study it was considered not only calcified plaques but also fibrolipid and mixed ones.
Conclusion: for me the important result of the paper is the fairly high frequency of this anomaly. The majority of patients do not go to coronary angiography after a coroscanner which seeks above all to rule out the disease. I'm not sure it can be said that, if you don't mention a double LAD, there will be inadequate theapetutiau. To review.
Thanks for the suggestion; although dual LAD is usually a benign variant and is not associated with a high frequency of angina, it is important to report it if it is has an aberrant origin and in order to better identify the site of the stenosis when present, in view of hemodynamic treatments and procedures.